# An Evaluation Framework for Urban Cadastral System Policy in Ethiopia

**Solomon Dargie Chekole [1],\*, Walter Timo de Vries [2]  and Gebeyehu Belay Shibeshi [1]**

[1]  Institute of Land Administration, Bahir Dar University, Bahir Dar 6000, Ethiopia; gebeyehu.belay@gmail.com
[2]  Department of Aerospace and Geodesy, Technical University of Munich, 80333 Munich, Germany; wt.de-vries@tum.de
\*  Correspondence: Solomondargie@yahoo.com

**Abstract:** Land is the most vital resource on earth from which people derive their basic needs. In order to administer and manage this vital resource in a sustainable way, there are several mechanisms, of which the cadastral system is the prime one. Literature documents that the performance measurement methods of cadastral systems are not appropriate. In most developing countries, systematic performance evaluation mechanisms for cadastral systems are very inadequate. For example, Ethiopia has no systematic evaluation framework to measure and evaluate the state of cadastral systems. This article aims to develop an evaluation framework to measure and evaluate the performance of urban cadastral systems in Ethiopia based on the methods that have proven successful in developed countries. The goal is furthermore to present a set of good practices and a set of indicators that can provide an objective basis to support a systematic evaluation of urban cadastral systems in Ethiopia. The study employs a desk review research strategy and a qualitative analytical approach.

**Keywords:** cadastral system; cadastral policy; performance indicator; evaluation framework

## 1. Introduction

Land is the most vital resource on earth from which humankind derives almost all its basic needs. Much effort is invested in order to administer land, of which cadastral systems are one of these efforts that are developed all over the world [1]. The United Nations and organizations such as the International Federation of Surveyors (FIG) have for many years undertaken studies to understand and describe land administration systems and particularly the cadastral system component [2].

For this paper, the term cadastral system is defined as a formal sub-system of land administration that includes the organizational system (a set of professional actors with accountable responsibilities to carry out cadastral activities and maintain cadastral information systems), procedures, and regulations, which altogether ensure that the cadastral system is kept up-to-date. In short, a cadastral system is an organizational system usually referring to the operations that a cadastral organization is conducting [3]. Urban cadastral system in this context refers to a cadastral system in an area where there is human settlement with high population density and infrastructure of built environment.

Urban cadastral systems are highly valuable for generating and distributing comprehensive data during land administration and management processes, and this can be regarded as a cornerstone for efficient operations of any state [4]. The establishment of urban cadastral systems in developing countries supports the provision of security of tenure by both ensuring that the information on urban tenure is formally acknowledged and by using the information as a basis for any planning decisions and interventions. Silva and Stubkjaer [5], deSoto [6], and deSoto [7] argue that the lack of a reliable

and efficient cadastral system can have serious implications for the social and economic welfare of a country.

A cadastral system constitutes the core of land administration functions, including the administration and management of land tenure, land value, land use, and land development. The land management paradigm of Enemark [8], presented in Figure 1, describes cadastral systems as the engine of any land administration system and posits that these underpin any country's capacity to deliver sustainable development. Cadastral systems produce cadastral information, which is required to make and implement different decisions on land. They encapsulate the location of land parcels, define boundaries, and provide parcel sizes. All of these information products are fundamental pieces of information for land allocation procedures, valuation of properties, and reduction of land-related conflicts and litigation.

Cadastral systems are not ends in themselves; rather, they are a means to support a variety of purposes [9]. Cadastral systems facilitate administration (i.e., both the information management as well as the regulatory processes) of three main areas: Land tenure, land use, and land value [8].

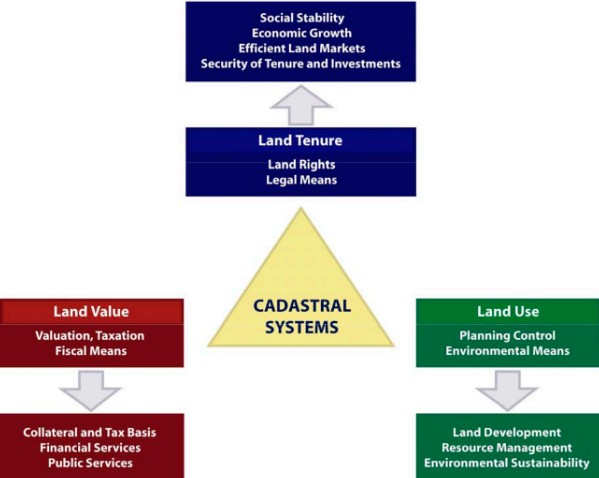

**Figure 1.** The nexus among cadastral system, land use, land value, and land tenure [8].

All these three functions are interrelated. Without appropriate cadastral systems, land tenure cannot be guaranteed, and controlling land use is not possible without up-to-date information and accurate updating mechanisms. This function supports land use management, the system of legal requirements, and regulations that apply to land in order to achieve desirable and harmonious development of the built environment. Land valuation and taxation are also the results of an appropriate cadastral system. The actual economic and physical use of land and properties influence land value. Land value is also influenced by the possible future use of land as determined through zoning, land use planning regulations, and permit granting processes, and the land use planning and policies will, of course, determine and regulate future land development.

There is growing interest internationally in cadastral systems and especially in their role as part of a national spatial data infrastructure (SDI). The role the cadastral system plays in supporting sustainable development is also well accepted [10]. Both developed and developing countries accept the need to evaluate cadastral systems to help identify areas of improvement and whether their systems are capable of addressing future needs.

The UN [11] has envisioned 17 goals for 2030 in the form of Sustainable Development Goals (SDG). The SDGs are the blueprint to achieve a better and more sustainable future for all. They address the global challenges we face, including those related to poverty, environmental degradation, prosperity, and peace and justice. According to Rajabifard [10], countries require access to an effective, efficient, and modern land administration system (LAS) based on a cadastre engine that contains spatially accurate land parcels and corresponding rights, restrictions, and responsibilities to achieve

sustainable development goals (SDGs). Some of the targets set in the SDG in relation to cadastral systems [10] are: Goal 1: No poverty—this goal guides implementation of nationally appropriate social protection systems (1.3) and creation of sound policy frameworks at the national, regional, and international levels (1.B); Goal 8: Decent work and economic growth; Goal 11: Sustainable cities and communities; Goal 16: Pease, justice, and strong institutions. In supporting and achieving these goals, cadastral systems play a significant role through securing societies' property ownership. The Bathurst declaration has confirmed the powerful link between appropriate cadastral systems and sustainable development [12]. The use of spatial information systems as key tools in national land management and meeting sustainable development objectives is growing, but every nation uses them in different ways. Operations of the technical aspects of cadastral systems are expensive in terms of financial and human resources. Sharing experiences and information is also not as simple as implemented in the developed countries unless we develop language of comparison and an evaluation framework [13].

In recent years, there has been an increasing interest in the use of evaluation and performance indicators by multilateral organizations to assess the outcomes of national projects and programs related to land. Performance assessment systems are not new to this domain. There is already a variety of frameworks and methodologies that can evaluate, characterize, and assess cadastral systems [14]. In order to determine the fulfillment of SDG objectives, cadastral systems need to be effective and sustainable in their implementation. For this reason, performance evaluation mechanisms are needed [15]. Standardized methods or a quality framework to measure and evaluate urban cadastral system around the world are still lacking [16]. This is maybe largely due to the fact that the natures of cadastral systems are dependent on the cultural and social values of the societies of the prevailing country in which they operate [17]. To date, research has primarily focused on the usage, principles, advantages, and disadvantages of the existing implementations of the urban cadastral system, yet the design of systematic evaluations of the systems is still insufficiently researched [18]. Although several authors have addressed evaluation frameworks for different aspects of land administration and management [19,20], a specific method evaluating urban cadastral systems seems to be missing. According to Gebrewold [21], the key difficulty for the success of cadastral systems is the absence of standardized frameworks, which enable evaluation of the performance of undertaking institutions. In this respect, for Ethiopia, as an implementer of two types of cadastral systems (urban and rural), there is no nationally accepted methodology that can measure and evaluate the performance of the urban cadastral system.

Therefore, the objective of this study is to develop a framework, with a set of good practices and their indicators, that can measure and evaluate the performance of urban cadastral system policy of Ethiopia so that an objective-based systematic evaluation of urban cadastral system can be made. In line with this objective, the research has endeavored to answer the question: With which indicators can one measure and evaluate the urban cadastral system of Ethiopia at policy level? Hence, the overall contribution behind evaluation of the cadastral system performance at policy level is to inform policymakers, practitioners, and stakeholders about the progress of the cadastral system projects in achieving their intended objectives.

The paper is structured through five main sections. The first introduces the concept and definition of cadastral systems. The subsequent section clarifies the theoretical knowledge and background of evaluation systems in general and evaluation frameworks related to cadastral systems in particular. This is followed by a methodology section, describing how data were collected and analyzed, followed by a results section describing and discussing how and to what extent new evaluation frameworks for cadastral systems can be derived. Finally, the paper concludes by answering the research question.

## 2. Theoretical Framework: An Evaluation Framework

Both the development of spatial information technologies and the drivers of sustainability have fostered the creation of new visions and models for cadastral systems. The 2030 Agenda for Sustainable Development is a global plan of action for people, planet, prosperity, peace, and partnership.

Considering that an estimated 70% of people do not enjoy secure land and property rights, there is a need to accelerate efforts to document, record, and recognize people-to-land relationships in all forms. The 2030 agenda includes 17 goals and 169 targets/indicators adopted on 25 September 2015 by heads of state and government at a special UN summit [11].

Gradually, the cadastral systems have been conceptualized by multipurpose cadastres [22], Cadastre 2014 [23], the Bogor Declaration on Cadastres [24], The Bathurst Declaration [12], and the development of sustainable land administration systems. How cadastral systems can support sustainable development is articulated in the Bogor Declaration and The Bathurst Declaration. The Bogor Declaration additionally formulated a way for cadastral systems to combat poverty and environmental decline. The Bathurst Declaration has also articulated the link between good land administration and cadastral systems; the cadastre enables good land administration by providing reliable and usable land information. Enemark et al. [8] formulated the so-called "Land Management Paradigm", portraying the link between a country's land policies, land administration functions (tenure, use, value, and development), information infrastructures, and the achievement of sustainable development.

A cadastral system can be seen as part of the operational level of land administration. It includes the information systems and processes to facilitate the application of land-related rules and regulations. It also supports the cadastral institutions in their mandate to maintain transparency, accountability, and responsiveness for the management of social, economic, and environmental issues [25]. Cadastral information systems are reliable and accurate datasets, which contain evidence for land-related transactions. The reliability and effectiveness of this evidence can be enhanced if there is a framework for performance evaluation of cadastral systems [26].

An evaluation framework is a systematic approach, which provides an assessment of the quality of current activities and of the system as a whole. This systematic approach identifies good practices and their indicators in order to assess the strengths and weaknesses of operational performance of an organizational system [16]. According to the UN declaration [27], under chapter 40 (4), one of the key underlying assumptions about indicators is that they are a precondition for improved decision-making [27]. According to international standards, a framework provides an evaluation technique that enables the identification of indicators for cadastral systems subject to improvement [21,27–29]. An indicator is a specific, observable, and measurable characteristic, which shows changes or the degree of progress of a system in achieving a specific outcome [20]. By understanding how urban cadastral systems function in countries where cadastral systems are effective, it is possible to derive a set of good practices and possible success factors [16]. Such an approach may be relevant for developing countries, which are struggling with finding solutions for institutional problems with cadastral information. By studying, comparing, and analyzing how other countries reach stability and sustainable outcomes in their cadastral systems, it is possible to identify the most relevant elements and aspects [30]. International literature shows that there are a number of evaluation frameworks that can measure and evaluate organizational performance. As stated by Gebrewold [21], good practices and indicators are reference points for evaluations, and they constitute a critical component of an evaluation framework.

In Ethiopia, many cadastral projects have been implemented, yet with varying degree of success. Each of these (pilot) projects contained trials for implementing cadastral systems, yet often these were not complementary to existence of earlier projects. This has resulted in overlaps, redundancies, and ill-functioning and inconsistent cadastral systems throughout the country. One of the most notable characteristics of these projects was the consistent absence of a progress performance evaluation of the project in each project phase. In other words, there is no systematic assessment and evaluation of the strengths and weaknesses of earlier projects, and there are no systematic set of guidelines used at the start of projects. On the other hand, there are standards and indicators developed by international organizations that can perform these tasks, yet these standards are not adopted. For instance, the FIG has proposed a set of criteria for the development and evaluation of cadastral systems [28].

In 2004, Steudler developed an evaluation framework for land administration consisting of policy, organizational management, operational, partnerships, and review processes. Furthermore, Shibeshi et al. [17] adapted Steudler's framework to evaluate the status of land administration systems in Ethiopia. Rajabifard [29] developed the so-called cadastral template, which was originally a way to compare cadastral systems globally, but gradually developed into a standardized system, which describes and evaluates the key elements of any cadastral system and has as such become a standard of designing and evaluating cadastral systems. Outside of the cadastral domain, there are evaluation frameworks to evaluate performances of organizational systems, such as the European Foundation for Quality Management (EFQM). This framework evaluates the effectiveness of an organization based on nine indicators. These indicators represent both the input and output of an organizational system. The input indicators are referred to as enablers (leadership, strategy, people, partnership, and process), whereas the output indicators are referred to as results (people, customer, society, and business). The model posits that the achievement of sustained success relies on strong leadership and a clear strategic direction. Further enablers are the need to develop and improve their people, partnerships, and (internal) processes. Only once these elements are in place is it possible to deliver value-adding services to customers. [28]. Another example of an evaluation framework is the "Land Governance Assessment Framework" (LGOV). It is an investigative instrument to assess the status of land governance in a particular country. With a set of strictly defined indicators this LGOV helps to compare a country's land governance to other countries. This can stimulate a particular country to adapt its land governance.

Development practitioners of all persuasions recognize that a well-functioning land sector can boost a country's economic growth, foster social development, shield the rights of vulnerable groups, and help with environmental protection [21]. The aforementioned frameworks are aimed at measuring and evaluating the performance of urban cadastral systems policy. Finally, an analytical framework has been developed to guide and facilitate understanding of cadastral systems.

In this article, we analyze peer-reviewed literature on the future visions and available models (see Table 1) on cadastral system, with the purpose of establishing a framework and a methodology that will help measure and evaluate the performance of national urban cadastral systems.

**Table 1.** Benchmark frameworks for urban cadastral system.

| Framework/Model | Author/s | Indicators | Good Practices |
|---|---|---|---|
| European Foundation for Quality Management (EFQM) Excellence Model | [28] | Enablers: Leadership, strategy, people, process, partnership; results: People, customer, society satisfaction and business results. | Excellent organizations have leaders (who shape the future and make it happen) and people (who strive for achievement of organization's goal) and develop a stakeholder-focused strategy. |
| Land Governance Assessment Framework | [21] | 1. Legal and institutional framework.<br>2. Land use planning, management, and taxation.<br>3. Management of public land.<br>4. Public provision of land information.<br>5. Dispute resolution and conflict management. | Regulations and management of land involve institutions with clear mandates as well as policy processes that are transparent and equitable. Processes for land use planning are efficient, and taxation on land is transparently and efficiently collected. Policy makers assess the extent to which public land holdings are justified and transparently inventoried and managed. Land information systems provide sufficient, relevant, and up-to-date data, at a cost affordable to the general public. Affordable, clearly defined, transparent, and unbiased mechanisms exist for the resolution of land disputes, and these mechanisms function effectively in practice. |
| Land Administration Evaluation Framework | [19] | Policy, legal, historical, financial, social, political, environmental, cost recovery. | There are policy documents, which consider integrated and multi-disciplinary aspects. |
| Land Administration Evaluation systems | [17] | Policy, institutional, operational, monitoring, and evaluation aspects. | The policy-level evaluation testifies if the system is well defined by objectives, if it responds to the needs of the society, and if it is equitable for all. |
| Cadastre 2014 | [23] | 1. Complete legal situation of land (public, private).<br>2. No separation between maps and registers.<br>3. Cadastral mapping will be replaced by modelling.<br>4. Manual cadastre will not be longer.<br>5. Private–public partnership is strengthened.<br>6. Cost recovery. | The good practice is when cadastral systems are subjected to be evaluated in light of these six elements. Private–public partnership should be strengthened so that technical aspects of cadastre will be performed in supervision with the public |
| Cadastral Template (six quantitative and two qualitative indicators) | [29] | Registration systems, population vs. parcel, strata units, percentage of parcels registered, surveyors and lawyers, surveyors vs. lawyers, educational bodies, educational reform issues. | There are objective indicators to compare and assess cadastral systems. |
| The 2030 Agenda for SDG | [11] | Indicator: 1.4.2. Proportion of total adult population with secure tenure rights to land, with legally recognized documentation and who perceive their rights to land as secure, by sex and by type of tenure. | Cadastral system policies support and contribute to the goals of Agenda 2030. |

## 3. Methodology

This study has adopted a combination of a desk review of international literature, a case study, and a document analysis of Ethiopian documents. The desk review focuses on exploring and looking into existing literature on indicators and good practices of cadastral systems. The goal of this desk study was to explore how to measure and evaluate the performance of urban cadastral system policy of Ethiopia. Secondary data sources such as books, journals, and conference proceedings served as input for the desk review. From the compilation, seven frameworks and models could be identified as useful and relevant. These constitute the EFQM Excellence Model [28], the Land Governance Assessment Framework [21], the Land Administration Evaluation Framework [23], the Land Administration Evaluation systems [17], Cadastre 2014 [23], the Cadastral template [29] and The 2030 Agenda for SDG [11]. The rationale for selecting these frameworks is that each of these are flexible, reliable, comprehensive, and attainable. Different scholars have developed an evaluation framework from the concept of organizational pyramids [31]. For instance, Steudler et al. [32], Mitchell et al. [33], and Yilmaz et al. [27] propose an evaluation framework based on five levels, i.e., policy level, management level, operational level, external factors, and review process. These levels are divided into evaluation aspects, which are particular parts within levels. For each aspect, good practices and their indicators are developed. Based on the above literature, this paper develops an evaluation framework for urban cadastral system at policy level.

The second research strategy used was case study methodology. This was employed in order to assess to what extent the framework would be dependent on or independent from its context. The focus of the evaluation of cadastral system cases was to see to what extent cadastral experts and administrators in Addis Ababa city's eight sub-cities were consistent in their actions and behavior. Respondents were selected based on purposive sampling technique with the rationale that the research question requires special expertise in the area of cadastral system.

Semi-structured interviews were conducted with each director in eight sub-cities (Yeka, Bole, Addis Ketema, Lafto, Lideta, Kolfe, Gulelle, and Kirkos) out of 10, in which sub-cities were selected on the basis of good performance progress. The semi-structured interviews were bounded with the theme of cadastral system at policy level. Some of the guiding questions focused on the presence or absence of existence of political will; inclusion of cadastral system in the constitution; identified visions and objectives; existence of stakeholder-focused strategy; existence of cadastral policy for standardization, involvement of private sectors, digital cadastral data lodgment, the need of 3D cadastral system, and existence of clear laws and directives; the nexus between cadastral system policy and environment.

Group discussions were also made with cadastral experts in order to crosscheck and validate the responses with administrators. Experts consisted of six professionals (two lawyers, two land administrators, and two surveyors) from each sub-city who were given eight guiding questions. The questions were composed of cadastral system in light of political will, policy, institutional, social, economic, environmental, technical, and public–private partnership aspects.

The third research strategy was document analysis. This is a form of qualitative research in which documents are interpreted to derive meaning. The policy documents that were analyzed included the proclamation no. 818/2014 (dealing with the urban cadastral system adjudication and registration), the proclamation no. 721/2011 (dealing with urban land acquisition modalities, urban land policy development, and management), the urban land adjudication, and registration regulation no. 323/2014 and 324/2014 and the standard no. 04 and 05/2015.

## 4. Result and Discussion

### 4.1. Urban Cadastral System Policy of Ethiopia

The current land policy in Ethiopia is multi-dimensional. It consists of a complex of socio-economic and legal prescriptions, which dictate how the land and the benefits from the land are to be used properly. Balance must be kept between the exploitation, utilization, and conservation of the land as a resource in order to obtain the necessary level of sustainable development [30]. Within this complexity

of rules, the cadastral system policy is one component. It sets the principles with which the formal land administration system is guided and implemented.

Ethiopia's urban cadastral system policy is enshrined and incorporated under the urban land development and management policy. This policy aspires to a system where urban land is served as a driving force for political, social, economic, and environmental transformation through an efficient and well-functioning cadastral system. To accomplish this vision, the federal government has formulated policies related to urban cadastral systems in order to modernize the system of land administration. The proclamation No. 818/2014 describes urban land holding adjudication and registration. Practically, it describes how to implement the urban cadastral system of the country [34]. In line with this proclamation, a set of regulations, directives, and manuals have been prepared. The overall objective of the scheme is to accelerate the socio-economic and environmental development of urban centers by ensuring land holders' security of holding and recognition of title to immovable property by certifying their right, restriction, and responsibility through adjudication and registration. In line with this proclamation, Reg. No. 323/2014 and 324/2014 were issued to enact the proclamation [35,36].

To implement proclamation No. 818/2014, the government of Ethiopia has incorporated the whole urban cadastral system processes in to its Growth and Transformation Plans (GTP). The Federal Urban Land and Land Related Registration and Information Agency, which is the responsible organ for all urban cadastral systems, has developed an aspiring agenda of urban land registration to support GTP II, which is the second strategic plan (2014–2019). Within this framework, adjudication and registration of 1.6 million and 1.2 million landholdings respectively across 91 cities are planned in five years with 200,000 adjudicated and 150,000 registered in just the first year across prioritized 23 cities [37].

*4.2. An Evaluation Framework for Urban Cadastral System Policy*

The proposed framework consists of four components: (administrative) level, which describes the government level at which guidelines for cadastral systems are formulated; aspect, which refers to the evaluation characteristics; indicators, which reflect the variables and proxies that can be measured and with which quality of the system can be described; and good practices, which reflect the degree to which the activities contribute to successful achievements. The focus of this paper covers urban cadastral system at policy level, excluding management and operational levels in the "organizational pyramid" of government. In this respect, policy level is the higher hierarchy, where a set of principles guide decisions and achieve rational outcomes. As explained in Table 2, different aspects are stated under the policy level that need to be incorporated: Political aspects, policy aspects, legal and institutional aspects, social aspects, economical aspects, environmental aspects, technical aspects, and public–private partnership aspects. Performance indicators are measurable values, which demonstrate how an organization is obtaining targeted objectives. Organizations use performance indicators to evaluate the degree of success in relation to the objectives. By understanding how a cadastral system can be efficiently implemented and maintained, it is possible to define the good practices and the success factors in terms of different aspects [19].

*4.3. Urban Cadastral System Policy Evaluation Aspects*

Currently, there is an increasing demand for cadastral systems, whose contribution to economic development, environmental management, and social stability is more visible and measurable [9]. In line with this changing environment, there is a stronger need than ever for performance measurement, which can demonstrate and ultimately assure the quality of cadastral system. In other words, at the cadastral system policy level, eight elements are identified as aspects in which performance indicators are contained within. These aspects listed in Table 2 form the basis for the evaluation of urban cadastral system policy performance. The details of these aspects are explained in the following section.

### 4.3.1. Political Aspects

A functional cadastral system is the foundation for political stability, social welfare, economic development, and environmental protection. A government can develop a comprehensive vision to boost the country's economy, but without its political will and commitment, the authorities will never be able to deliver. The cadastral system success is driven by the level of political will and commitment, which take into account the social, economic, and cultural contexts and makes the necessary resources available [38]. A well-functioning urban cadastral system can never be achieved without positive political will. Although all aspects (Table 2) have their own contribution to the successful implementation of a cadastral system, Enemark [8], has highlighted most importantly strong political will and leadership as a fundamental requirement at national level. Without the courtesy of strong political will and commitment, a cadastral system would not be successful. The good practice is thus, political institutions of the country under investigation should demonstrate their will and commitment to accomplish the objectives envisioned.

From the interview and discussion made, cadastral system administrators have noted the need for continued support and political awareness of the benefits of an effective and efficient cadastral system. In addition, the overarching policy required to capture the key principles are essential for establishing effective, efficient, sustainable, and interoperable cadastral systems. They have also noted that the principles-based overarching policy guidance should be flexible, recognizing the diverse social and economic contexts within national and regional cadastral system arrangements. In doing so, close collaboration between central and regional bodies should be mandatory in order to avoid any gap and duplication. All in all, it can be concluded from the interview and group discussions that political will is the determinant factor (indicator) during the development and implementation of cadastral system.

### 4.3.2. Policy, Legal, and Institutional Aspects

A system for recording land ownership and other land-related data is an indispensable tool for a market economy to work properly, as well as for sustainable management of land resources. Cadastral system policies could include principles on the roles and responsibilities of the various cadastral-related activities such as land surveying, mapping, and land registration. These principles could be included in the national cadastral system policy. In line with this, there should be a strategic tool to know how to achieve mission, vision, and targets. Excellent organizations implement their mission and vision by developing a stakeholder-focused strategy. These strategies should be designed in a SMART way, i.e., Specific: Objectives are concrete, detailed, focused, well-defined, straightforward, and emphasize action; Measurable: The standard used for comparison, it answers the question of quantity; Attainable: Objectives need to be realistic, possible, and achievable; Realistic: What results can realistically be achieved, given available resources; Time-bound: The deadlines to meet the objectives.

The federal government of Ethiopia has issued a proclamation and the respective subordinate laws to ensure that the boundaries of real property are accurately marked, measured, and mapped. On the contrary, inadequate policy formulation and implementation will hinder the functioning of cadastral systems. In this case, the good practice is the provision of legal recognition through enabling legislation that covers all the details and standard procedures of the processes. In addition, it is good when the legal aspects are suitable to the cadastral system through protecting ownership rights that people have on land and property. Although dependent on policy and legal aspects, inappropriate institutional arrangements are often a severe limitation in any cadastral system [39], so it is important to combine all of the different cadastral system activities under the control of one specific state department, though the decentralization of the activities and functions to regional level is likewise important. In support of this issue, Williamson [22] has proved that the most successful cadastral systems have been established as a result of all cadastral system activities being combined into one government agency.

In support of the institutional arrangement of cadastral system, UN-GGIM [40] has affirmed in the Addis Ababa declaration for "Good Land Governance for Agenda 2030" that strong land administration institutions are required to support effective and efficient land administration and

management to address the need to secure land and property rights for all. According to the view by cadastral system experts and administrators, incorporation of strong institutional arrangements in the proposed framework makes it sound for the organizations' performance evaluation. This type of arrangement will help applicability of uniform cadastral system policy throughout the country. Since regulations and directives are enacted in accordance with this policy, there will be institutions with clear mandates as well as processes.

**Table 2.** An evaluation framework for urban cadastral system.

| Level | Aspects | Performance Indicators | Good Practices |
|---|---|---|---|
| Policy level | Political | Existence of political will in support of the cadastral system (y/n). | When there is clear political will to advance cadastral system of the country. |
| | Policy | Existence of a government policy for cadastral system (y/n). Are the identified visions and objectives SMART? (y/n) Existence stakeholder-focused strategy (y/n), if yes, what is the strategic approach that has been adopted to meet the objectives? Frequency of revisiting objectives and strategies Existence of cadastral policy for: <br>■ Supporting Agenda 2030 for SDG: Indicator 1.4.2 * (y/n) <br>■ Digital cadastral data lodgement portal (y/n) <br>■ High speed internet for digital data lodgement (y/n) <br>■ Developing in 3D digital cadastral system (y/n) <br>■ Data preparation, sharing, IP, etc. (y/n) <br>■ Base-map preparation and maintenance (y/n) | When cadastral policy aspects are mentioned in the land policy and are suitable to circumstances. Specific, measurable, achievable, realistic and timely (SMART) There should be a strategic tool to know how to achieve mission, vision, and targets. Excellent organizations implement their mission and vision by developing a stakeholder-focused strategy. Plans, objectives, and processes are developed and deployed to deliver the strategy [28]. When there are progress-monitoring mechanisms on the basis of objectivity. When the cadastral system policy supports and contributes to the achievement of SDGs. When there is a lodgement portal for loading digital cadastral information. When there is special internet designed for this purpose. While the policy fulfilling the current needs, there should be possibility for developing 3D cadastre. When the cadastral policy guides data preparation, sharing, IP, etc. When there is base-map preparation and updating. |
| | Legal and institutional | Existence of legal basis, such as laws, regulations, standards (y/n). Uniformity of cadastral system policy throughout the country. Do regulations of cadastral system involve institutions with clear mandates as well as policy processes that are transparent and equitable? (y/n) if yes, explain. | Legal recognition through enabling legislation that covers all the details and standard procedures. When the legal aspects are suitable to cadastral system through protecting ownership rights that people have on land and property. When there are institutions with clear responsibilities and easy processes in the cadastral system [21]. |
| | Social | Does ensured participation in the cadastral system lead to policy development, such as stakeholders? (y/n) if not, why? Does the society benefit from and acknowledge the policy? Is there any mechanism for resolving disputes arising among landholders? | When participation is ensured. Implementations are possible when public–private sector partnership cooperates and increases achievability of missions and objectives. Society should benefit from and acknowledge the need of the cadastral system policy. Good practice is when there are hierarchical dispute resolution mechanisms (negotiation, arbitration) |

**Table 2.** *Cont.*

| Level | Aspects | Performance Indicators | Good Practices |
|---|---|---|---|
| Policy level | Economical | Is there a cadastral system policy for cost recovery? (y/n) if yes, how and in what mechanisms? Is there a well-functioning land and property market as a result of the cadastral system policy? (y/n) | Cadastral system procedure should be self-financial and should ensure cost recovery. Cadastral system policy should support a well-functioning land market. |
| | Environmental | Does the cadastral system policy ensure sustainability of the environment? (y/n), if yes, in what aspects? | Cadastral system policy needs to support duties such as environmental protection, monitoring of land resources, zoning, etc. |
| | Public–private partnership | Does the system encourage involvement of the private sector? (y/n), if not, why? Does the policy encourage commercialization of registration? (y/n) | Private sector is the indispensable partner of the public sector in terms of its capability in using and adjusting modern technologies. Thus, a good practice is when there is partnership of public and private sectors under the condition of well-determined limits of both parts' duties and responsibilities [23]. |
| | Technical | Does the cadastral system policy follow international technical standards? (y/n) if not, why? Existence of international standards such as LADM, technical standards (y/n). | When cadastral systems follow international standards so as to share information, taking into account international standards [41]. When the cadastral policy adopts and customizes international technical standards. |

\* Agenda 2030 SDG: Goal 1; indicator: 1.4.2. Proportion of total adult population with secure tenure rights to land, with legally recognized documentation and who perceive their rights to land as secure, by sex and by type of tenure.

The first goal of Agenda 2030 is "End poverty in all its forms everywhere". Under this goal, indicator 1.4.2 stated as "Proportion of total adult population with secure tenure rights to land, with legally recognized documentation and who perceive their rights to land as secure, by sex and by type of tenure". This indicator directly points to the cadastral system. Thus, in order to support and contribute to the achievement of SDGs, the role played by the cadastral system will be significant through documentation, registration, and recognizing people-to-land relationships. In doing so, indicators related to land administration and management need to be articulated in the cadastral system policy.

Other elements of the cadastral system, such as a digital cadastral information lodgment portal that enables to reposit digital information; provision of high speed internet in order to facilitate uploading information; data preparation, sharing, IP, etc.; mechanism of base-map preparation; and maintenance need to be incorporated in the cadastral system policy. Technological advancements will enable the existing 2D cadastre to be extended to the 3D to incorporate height into cadastral frameworks. However, the current context and circumstances of Ethiopia do not provide this opportunity for different reasons. The economic capacity, policy, and legal barriers hinder the development and operations of a 3D cadastre. Thus, the 2D cadastral system can be developed through an incremental approach as per the principle of fit-for-purpose.

### 4.3.3. Social, Economic, and Environmental Aspects

A cadastral system provides order and stability in a society by creating security not only for landowners but also for investors and moneylenders and for governments. Although systems of land registration are frequently directed at protecting the interests of individual landowners, they are also instruments of national land policy and mechanisms to support economic development [30].

An urban cadastral system policy plays a significant role in improving and boosting the social and economic status of a country. In terms of this social dimension, a well-functioning cadastral system is the foundation of national stability and social welfare. A government can make a thousand promises

or grandly announce a comprehensive vision to boost the country's economy, but without an efficient and effective cadastral system the government will never be able to deliver.

A cadastral system offers countries a means of escape from poverty, as they secure land tenure and provide stability in the land market [42]. As economic development is one of the common goals of many developing countries such as Ethiopia, one could argue that the current policy issued by the Ethiopian government to implement urban cadastral system advances the level of economic development. In addition, one could posit that cadastral systems enable the translation and implementation of social policies if they enable fair registration of people of different sociocultural groups and gender. In line with such social and economic aspects, a cadastral system policy needs to support governance tasks such as environmental protection, monitoring of land resources, and land use zoning. A cadastral system can also be used for the preparation of environmental impact assessments and for monitoring the impacts of development projects. Hence, when evaluating and describing the economic status of a country, it is wise to zoom in on the progress of its cadastral system policy and evaluation.

The main outcome from the improvement of the cadastral system is serving the community through securing its tenure [15]. With this in mind, the cadastral system policy needs to be considered and assured of participation of concerned bodies such as stakeholders, professional associations, civil servants, etc. during the development of cadastral system policies. This issue has been suggested by the group of cadastral experts and administrators in order to be able to develop a comprehensive framework. In contrast to the results of Steudler et al. [19], Mitchell et al. [33], and Yilmaz et al. [27], which all limit the applications and impacts of cadastral systems to legal and technical aspects in particular, it is recommended to expand the effects of cadastral systems beyond this narrow focus. Instead, their performance should also be connected to social, economic, and environmental concerns, since the effects also help communities by solving boundary-related disputes arising among landholders. All these issues should be part of the policy that can be served as a benchmark during evaluation phases.

### 4.3.4. Public–Private Partnership (PPP) and Technical Aspects

The private sector is the part of the economy that is run by individuals and companies for profit and is not state controlled. The private sector is the indispensable partner of the public sector in terms of its capability in using and adjusting modern technologies. Currently, in most developed countries such as Switzerland and Australia, the technical part of cadastral systems (adjudication, boundary demarcation, registration, and related) is commercialized to the private sector. This has been experienced in many countries and has become successful. Ways of involving the private sector should be evaluated. Many countries apply legislation under which field surveys are undertaken by private licensed surveyors. Databases can physically be operated by private data centers, subcontracted by the relevant public authority [30]. By doing so, commercialization brings new services to market for the betterment of income from land. In this respect, de Soto [7] described "the impact of cadastral system commercialization results in faster project completion and reduced delays on infrastructure projects by including time-to-completion as a measure of performance and therefore of profit". According to FIG [23], strengthening and cooperating with the private sector helps in recovering the cost of the investment in the land. Thus, a good practice is when there is partnership of public and private sectors under the condition of well-determined limits of both parts' duties and responsibilities [23].

A good land information system includes textual files and spatial information that are closely linked to each other. In some of the countries, field surveys are undertaken by private surveyors, in other countries by governmental or local public agencies. The requirement for geometric precision varies considerably. Some countries require very precise surveying and mapping of boundaries, whilst others are far less demanding in this respect [30]. The surveying and mapping, which should be performed by the private sector, rely on high amounts of resources. The technical solution to address this problem would be more user-driven land information systems, yet a lack of technical standards, PPP, and the ability to share land information currently still hinder a sufficient performance of cadastral systems. This can be improved by a better alignment of the computerization of cadastral information



systems to the actual purpose of the cadastral systems, namely to provide legitimacy of ownership within society. The provided land information should not only adhere to acknowledged technical standards through which land information can be connected to other types of information, but it should also fit in other government processes, so incorporating these issues in the cadastral system policy will support the other evaluation aspects.

## 5. Conclusions and Recommendation

In conclusion, the purpose of this study was to develop a methodology to measure and evaluate the performance of urban cadastral system based on International Scientific Indexing (ISI) published journals and cadastral models. Finally, the paper contributes and develops an evaluation framework for urban cadastral system policy. The framework defines good practices and their indicators of an ideal urban cadastral system application. This contribution could be used for systematic evaluation and comparison of cadastral system practices. This paper proposes that evaluating urban cadastral systems policies should be connected to broader economic and societal issues. Such an evaluation framework should consider the combination of political, legal and institutional, social, economic, environmental, technical, and public–private partnership aspects. The framework provides a basis for evaluating urban cadastral systems policy in a more standardized and comprehensive approach. All in all, the result of this paper will enable policy makers, management officials, and implementers to follow up, monitor, and evaluate the strengths and weaknesses of the cadastral system performance in response to improvements in an organizational capacity, technology, and availability of access to spatial information. Thus, based on the result obtained, the researchers suggest the following recommendations:

1. In legislating and implementing an urban cadastral system, the issues of political willingness and commitment, legal and institutional issues, socio-economic aspects, environmental influences, and technical standards should be at the heart of the urban cadastral system policy.
2. Decision makers, key politicians, and professionals should be involved in selecting the best system for their country as their support is essential to the establishment of sound cadastral policies and the creation of an appropriate land administration system.
3. While the ultimate responsibility for the urban cadastral system lies with the government, the private sector may have a significant role to play in the cadastral system policy implementation, so the cadastral system should be underpinned by effective partnerships and co-operation between public and private sectors and the end user communities.
4. Any organization carrying out cadastral system implementations should follow the procedures in measuring and evaluating performance of the operations in a timely manner so that their organizational excellence can be defined.
5. Decision makers, who are in a different hierarchy of policy making, should address and incorporate the issues of follow-ups, monitoring, evaluation, and controlling mechanisms in their organization.
6. Urban cadastral system implementing institutions should use the proposed evaluation framework as a benchmark rather than merely evaluating through annual report.

**Author Contributions:** Conceptualization, S.D.C., W.T.d.V., G.B.S.; methodology S.D.C.; validation, S.D.C., W.T.d.V. and G.B.S.; formal analysis, S.D.C.; investigation, S.D.C.; resources, S.D.C.; data curation, S.D.C.; writing—original draft preparation, S.D.C.; writing—review and editing S.D.C, W.T.d.V. and G.B.S.; visualization, S.D.C.; supervision, W.T.d.V. and G.B.S.; funding acquisition, S.D.C.", All authors have read and agreed to the published version of the manuscript.

**Funding:** This research was funded by the German Academic Exchange Service (DAAD). The APC was funded by DAAD.

**Conflicts of Interest:** The authors declare no conflict of interest.

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
