# Peer review of "An Evaluation Framework for Urban Cadastral System Policy in Ethiopia"

_land, doi:10.3390/land9020060_

Round 1

Reviewer 1 Report

This paper provides an evaluation framework for urban cadastral system policy in Ethiopia. The topic is rather interesting and deserves to be investigated. The article does contain all the components a research article is expected to (Introduction, Theory, Methodology, Results, and Discussion). There is, however, a lot of improvement to be done in every component before the manuscript should be considered for a publication. In short, the paper is lacking connection between theory and findings and the methods are not reported up to a standard of an academic article. In their current form, the findings of the study are too ambiguous and hard to interpret in light of the existing knowledge. The choice of framework elements (dimensions, indicators, and good practices) is too arbitrary and makes the reader wonder how the authors ended up with the proposed framework. In addition, the manuscript would benefit from a close editing.

First, I list the elements that are missing completely or that are reported inadequately in the manuscript. After that, some more detailed comments about the manuscript are given.

Missing or inadequately reported elements:

A paragraph of the structure of the article in Introduction A comprehensive review of the previous literature (Section 2) Metatext in the beginning of all sections (to help the reader to understand the connections between sections) Contribution of the study is not stated clearly Limitations of the study are not discussed at all Reflection of the results to previous studies is missing

Comments about the content:

Definition of an urban cadastral system is missing – considering that the research gap and research objective are identified using this concept, it is vital to provide the definition in the Introduction. In the Introduction, there is a part where you are talking about a methodology that can measure both urban and rural cadastral system – this is confusing because the title and everything else before that part indicates that the study focuses on the urban cadastral system. Section 2: First paragraph falls under methodology, overall the content is too generic and does not provide an image that authors master the related literature. I would suggest you clearly separate general evaluation frameworks and frameworks of land administration in the text, with emphasis on the latter ones. Title of the first table is misleading since it provides an overview of the benchmarked frameworks, not a new analytical framework for urban cadastral system. In addition, for many of the frameworks, the mentioned indicators are not indicators at all, but goals (e.g. Cadastre 2014). Also, it is unclear why some parts of the table are underlined. Description of the methodology is too vague and hard to understand: Are the benchmarked frameworks result of the review? How did you outline the literary sources? Without any explanation the choice of sources seems purposeful. Section 4.1. is a case description rather than results of any analysis. Policy levels of the proposed framework appear ‘out of nowhere’, and there is some confusion with the use of terms level and dimension. The choice of indicators and good practices is also arbitrary – there should be a method to back up your analysis. Section 4.4.: Land tenure, land value, and land use are not systems but functions (see reference 28, p. 123).

Author Response

First and foremost, the authors would like to thank the editor for giving us the chance to respond to the review. The authors also acknowledge reviewers for their constructive and substantive comments for making our manuscript more comprehensible and interesting. We have studied reviewers’ comments carefully and made a correction which we hope will meet your approval.

Reviewer 2 Report

This article aims to develop an evaluation framework to measure and evaluate the performance of urban cadastral system in Ethiopia at policy level. The article is in line with the aims of Land Journal. However, it is missing some fundamental parts including assessment of the proposed evaluation framework in Ethiopia and publishing the land registry’s feedback, as well as incorporation of the United Nations Sustainable Development Goals (SDGs) into the proposed evaluation framework. The manuscript is full of English grammar errors. Both UK and US languages are used in the manuscript. Check the whole manuscript in terms of English grammar errors and correct them. Both UK and US languages are used in the manuscript. Follow the journal instructions and use the only language recommended by the journal.

Abstract:

Line 10, page 1: ‘Many literatures have proved that the role of cadastral systems have been functioning as an engine for sustainable land administration system though their systematic performance evaluation mechanisms seemed to be poor.’ This research problem needs to be reworded as it has grammar errors and avoids the problem from being clear to readers.

Lines 13-14, page 1: ‘This article is aimed to develop an evaluation framework to measure and evaluate the performance of urban cadastral system at policy level.’ Describe the other levels that could be considered for evaluation purposes in addition to policy level.

Line 16, page 1: grammar error: replace ‘evaluates’ with ‘evaluate’

Lines 17-18, page 1: ‘In order to achieve this aim, the study has employed a desk review research strategy being qualitative approach is at the heart of the analysis.’ This sentence does not make sense because of grammar errors and needs to be reworded.

Line 19, page 1: suggest replacing ‘contributed’ with ‘developed’

Line 22, page 1: keywords: ‘indicator’ should start with capital letter. Semicolon after the last keyword should be removed.

Introduction

Line 28, page 1: grammar error: replace ‘administration land’ with ‘administer land’

Lines 39, page 1: it seems like there’s a need for a full stop after ‘own land’. Also, it’s not clear who ‘argues that …’. If the authors mean the researchers from references 4-6 argue something, the names of the authors should be written in the manuscript. For example, Silva and Stubkjaer [4] argue that ….

Lines 44-44, page 1: Authors have referred to some research in 2005, 2007 and 2010 and claimed that the world still lacks a standardized method to evaluate urban cadastral systems. However, we’re in year 2019 and there’s a need for referencing to some recent research work that can prove this claim.

Lines 49-50, page 2, grammar error: ‘... which enable evaluate ...’

Line 50, page 2: add ‘of’ between ‘types’ and ‘cadastral’

Introduction lacks the structure of the manuscript.

Theoretical Framework: An Evaluation Framework

Line 60, page 2: tow ‘which’ are used in the sentence. Reword the sentence.

Line 62, page 2, ‘According to international standards, …’ Authors should explain what international standards they are referring to.

Line 67, page 2: grammar error: replace ‘helps’ with ‘help’

Line 68, page 2: ‘As stated by [15]’. Referencing should be corrected. Name of the authors should be mentioned.

Line 72, page 2: grammar error: ‘which are not complementary each other’

Line 76, page 2: use ‘FIG’ instead of ‘International Federation of Surveyors’, since it has been defined earlier in the manuscript.

Line 78, page 2: Wrong referencing of ‘Shibeshi, et al. [18]’. Correct it to ‘Shibeshi et al. [18]’. Also, replace ‘has’ with ‘have’.

Why is the name of table 21 instead of 1?

There is no link between Table 1 and the text above it. Table name must be mentioned within the text close to that table.

Table 1, grammar error: ‘When policy documents are consider those aspects’ and ‘When there exists an objective based indicators’

The format of text in Table 1 differs for some items. Why are some text underlined? Make them consistent.

What are the 6 quantitative and 2 qualitative indicators for ‘Cadastral Template’ in Table 1?

Methodology

The authors should consider the assessment of their proposed evaluation framework too.

Result and Discussion

Line 115, page 4: GTP II should be explained.

Line 121, page 4: grammar error: replace ‘needs’ with ‘need’

Line 130, page 4: Do you mean Table 4.1? Anyways, table numbering is wrong. Follow the journal’s instructions.

Table 4.1: grammar error: ‘Does the cadastral system policy follows international technical standards? (y/n), if not why?’, replace ‘follows’ with ‘follow’. ‘Good practice is when there is an hierarchical dispute resolution mechanisms’, replace ‘an’ with ‘a’.

Line 163, page 7: The authors have referred to ‘Williamson’ but there’s no citation.

Line 184, page 7: referencing error: 28 [28]

Line 187, page 7: grammar error: replace ‘gives’ with ‘give’

In Technical aspect, authors have only considered international standards. However, the new robust infrastructure such as high-speed internet as well as technologies for developing new cadastral systems, services and databases eg a digital cadastral plan lodgement portal or a GIS application that can analyse the cadastral data should be also considered. Also, supporting 3D digital cadastral data is nowadays one of the key requirements for any cadastral system, which is missing in this evaluation framework.

‘Data’ aspect has not been considered in the evaluation framework. At the policy level, the framework should define policies for cadastral data preparation, sharing, IP, etc. cadastral map base preparation and maintenance is one of the critical indicators for evaluating a cadastral system.

The authors need to review the United Nations’ Sustainable Development Goals (SDGs) and incorporate them into the evaluation framework for Ethiopia. There are a few research papers on the relationship between SDGs and cadastral systems, and how cadastral systems should be improved to achieve SDGs for different jurisdictions.

Women’s right of land ownership as well as the equal access of men and women to land should be considered in the evaluation framework. Land tenure and valuation should be also part of the evaluation framework.

The authors need to briefly discuss about the impact of commercialisation of land registries, which is currently happening around the world, on cadastral systems, where they talk about public-private partnership.

The authors need to evaluate the cadastral system in Ethiopia using their recommended evaluation framework, then assess the outcomes with the land registry managers and add the results including framework proposed enhancements to this manuscript.

It’s not clear why the authors put section 4.4 in section 4, as it should be part of literature review and not their own work.

Conclusion and Recommendation

Line 203, page 8: grammar error: ‘Ethiopia has no a benchmark’

Authors have claimed that ‘It is believed that the framework provides a basis for evaluating urban cadastral systems policy in a more standardized and comprehensive approach’. This assumption needs to be tested in Ethiopia land registry and the results be published in this article. Otherwise, recommending an untested evaluation framework, which is not approved by land registry, to organizations and decision makers does not make any sense and is not logical.

Author Response

(The authors gave the same response as above.)

Reviewer 3 Report

some, short, subchapters could be grouped, while changing the title

Author Response

(The authors gave the same response as above.)

Round 2

Reviewer 1 Report

The authors have addressed the comments of the first review adequately and the quality of the manuscript has improved substantially. Following issues should be considered:

Grammar check to improve the readability of the paper Section 1, lines 64-67, you state that “the role cadastral system plays in supporting sustainable development is also well accepted”. This statement would be more credible if some references were added. Introduction, line 71, Is there a reference for the SDG’s related to cadastral system? Section 3, lines 197-200, The 2030 Agenda for SDG missing from benchmarked frameworks (based on Table 1) Section 3, When were the interviews and group discussions implemented? (Considering that in the previous version of the manuscript these data sources were not mentioned.) Section 3, lines 229-231, not part of methodology. Perhaps more suitable to Conclusions section. Section 4, Sub-section 4.1. is a case description, why is it placed under Results?

Author Response

Dear reviewer,

Reviewer 2 Report

The paper has been significantly improved. However, the below items should be still considered:

Tables and figures captions should be corrected based on Journal’s instruction. The whole manuscript needs to be proofread by a professional proof-reader. Grammar errors still exist in the manuscript. Line 71, page 2, there’s no reference to support the SDGs which are found relevant to cadastral systems. I suggest referring to chapter 21 of book ‘Sustainable Development Goals Connectivity Dilemma (Open Access)’ Line 156, page 4, authors referred to 5 enablers but listed only 4 (leadership, strategy, people and process) Line 233, page 6, the sentence ‘Thus the study had been limited to incorporate two sub-cities from the study area due to time limitation’ makes an opposite meaning to what authors want to declare. Change it to : ‘The study could not incorporate two sub-cities from the study area due to time limitation’ Line 317, page 8, typo in UN-GGIM Section 4.3.4 is still called dimension instead of aspects Line 386, page 10, a sentences is duplicated: So, incorporating this issue in the cadastral system policy will support the other evaluation dimensions. So, incorporating these issues in the cadastral system policy will support the other evaluation aspects.

Author Response

Dear reviewr,
